# The PATIENT Approach: A New Bundle for the Management of Chronic Pain

**DOI:** 10.3390/jpm13111551

**Published:** 2023-10-29

**Authors:** Pasquale Buonanno, Annachiara Marra, Carmine Iacovazzo, Maria Vargas, Serena Nappi, Francesco Squillacioti, Andrea Uriel de Siena, Giuseppe Servillo

**Affiliations:** Department of Neuroscience, Reproductive Science and Odontostomatological Science, University of Naples “Federico II”, Via Sergio Pansini, 5, 80131 Naples, Italy; dottmarraannachiara@gmail.com (A.M.); iacovazzo@tin.it (C.I.); vargas.maria82@gmail.com (M.V.); serena.nappi94@gmail.com (S.N.); squillacioti.f@gmail.com (F.S.); andreauriel@outlook.it (A.U.d.S.);

**Keywords:** chronic pain, management, assessment, education, minimally invasive treatments, pharmacological therapy

## Abstract

Background: Chronic pain is one of the most challenging diseases for physicians as its etiology and manifestations can be extremely varied. Many guidelines have been published and many therapeutic options are nowadays available for the different types of pain. Given the enormous amount of information that healthcare providers must handle, it is not always simple to keep in mind all the phases and strategies to manage pain. We here present the acronym PATIENT (P: patient’s perception; A: assessment; T: tailored approach; I: iterative evaluation; E: education; N: non-pharmacological approach; T: team), a bundle which can help to summarize all the steps to follow in the management of chronic pain. Methods: We performed a PubMed search with a list of terms specific for every issue of the bundle; only English articles were considered. Results: We analyzed the literature investigating these topics to provide an overview of the available data on each bundle’s issue; their synthesis lead to an algorithm which may allow healthcare providers to undertake every step of a patient’s evaluation and management. Discussion: Pain management is very complex; our PATIENT bundle could be a guide to clinicians to optimize a patient’s evaluation and treatment.

## 1. Introduction

Chronic pain affects about 30% of the adult population and it represents a crucial global health problem with a dramatic impact on socioeconomic systems [1]. Patients affected by chronic pain can experience a decline in quality of life, physical function, productivity, mental health, and social interaction [2]. Even though many recommendations have been published and several pharmacological and minimally invasive treatments are available, chronic pain continues to be overlooked and millions of people are still under-managed [3].

Pain management is one of the most difficult challenges for physicians as many factors can influence the way every patient experiences pain and reports his symptoms; moreover, different mechanisms can be involved in pain pathophysiology and they have to be carefully considered during a patient’s assessment and before choosing the adequate treatment among all the pharmacological and minimally invasive options so far available [4]. In addition, it is fundamental to periodically evaluate symptoms (as pain could change its characteristics over time), and to establish treatment efficacy in order to tailor the therapy to the patient’s needs and responses to it. Given the complexity of pain etiology and management, a multidisciplinary approach is mandatory and the healthcare team should have a different composition according to pain etiology and the patient’s characteristics.

In this narrative review, we tried to summarize the main evidence-based findings about the management of chronic pain in order to find a simple method to sequentially identify all the topics that must be focused on. To date, many articles have been published about the different aspects of chronic pain management, focusing their attention on specific themes; on the other hand, the literature is still lacking a work which concisely gathers all the information so far available into a comprehensive review. We coined the original acronym PATIENT: every letter of this acronym stands for a fundamental issue related to chronic pain management that clinicians should keep in mind to have a global approach to this condition. Our PATIENT bundle includes Patient’s perception, Assessment, Tailored approach, Iterative evaluation, Education, No pharmacological approach, and Team (Figure 1). We reviewed the core evidence and features behind each individual component of the bundle.

## 2. Materials and Methods

We performed searches in PubMed, ISI, Scopus, and Cochrane library databases to find the most relevant articles about every topic discussed in our narrative review from the inception to September 2022; we included all the types of articles in English, giving the priority to the most recent meta-analyses, systematic reviews, and randomized control trials. We used different search terms and their combinations for the different topics of the review: Appendix A reports the search strings used for every letter of the PATIENT acronym, the number of articles which were found, and the number of articles included in our work.

## 3. PATIENT Approach

### 3.1. Patient’s Perception

Pain is considered the fifth vital sign and the primary symptom that induces people to ask for medical assistance [5]. Many factors can alter nociception and influence the pain experience, so the same stimulus can be perceived in different ways by different subjects.

Advanced age is associated with an increased number of pain sites, higher pain-related disability, and increased pain threshold for low-intensity stimuli, which are differently processed in the insula and somatosensory cortex [6,7]. Older people experience a loss of exteroceptive function mediated by C-fibers, so they are at higher risk of burns, injuries, and bruises. Lautenbacher et al. introduced the concept of “presbyalgos” to underline the reduction in pain sensitivity with age; the pain threshold, for some types of stimuli such as pressure or electricity, seems to not be influenced by age because deep tissue nociceptors and the direct activation of A-fibers are not impaired. Regarding pain tolerance (i.e., the maximum stimulus a subject can bear), although correlated with pain threshold (i.e., the minimum intensity of a stimulus experienced as painful), it showed no change with age; this could be explained by a faster rise in the perception of pain intensity as the energy of the stimulus increases [7]. 

Sex-related differences in pain perception have been deeply investigated, but no consistent conclusions can be drawn. In contrast with previous research showing that females had a lower pain threshold and tolerance, Racine et al. underlined that such differences depend on the type of stimuli: females tolerate less pressure and thermal pain (both cold and heat) than males, but no differences have been recorded in ischemic and cancer pain [8,9,10,11]. In addition, psychosocial factors and personal history could differently alter pain perception [12]. The gender role expectations (i.e., males are by convention supposed to be stronger than females) and the masculinity–femininity traits may explain sex-related and interindividual differences in perceived and reported pain [12,13]. As found by Fillingim et al., women who reported multiple distressing episodes in their life have lower pain threshold and tolerance, while no differences were observed in men; moreover, the frequency of different types of pain recorded in the same household and childhood sexual abuse may adversely affect pain perception, especially in women [13,14].

It is well-documented that negative moods (e.g., anxiety, anger, depression) are associated with more significant acute and chronic pain levels, so their management is fundamental. Moreover, coping strategies vary across individuals and are related to the different clinical manifestation of pain symptoms [15]. 

Stroemel-Scheder et al. showed that sleep deprivation leads to hyperalgesia, but pain sensitivity changes are reversed with the restoration of sleep [16]. This occurs more promptly following short-term sleep disturbances, whereas a longer period of sleep disorders may delay the recovery after sleep normalization. Changes in sleep architecture during aging (e.g., lower sleep efficiency, less slow-wave sleep) may make older subjects less susceptible to the effects of sleep disturbances [16].

Body weight seems to affect pain perception; in fact, researchers reported a higher pain threshold in obese subjects [17]. Price et al. observed an increased sensitivity to painful stimuli in areas with an excess of subcutaneous fat (e.g., abdomen), while no significant differences were recorded in areas with less subcutaneous fat (e.g., forehand and hand), probably due to local mechanical and chemical factors related to the excess of adipose tissue such as decreased density and increased depth of nerve fibers, and higher levels of anti-inflammatory cytokines [18]. Conversely, other data showed that body weight loss after bariatric surgery does not alter the higher pain threshold in obese patients, suggesting that other factors are likely involved in the modulation of pain pathways in obese patients [18].

Due to the extreme inter- and intra-individual variability in pain perception, physicians should always believe patients’ evaluation of their own symptoms, avoiding minimizing their pain report. Even when clinical or instrumental tests are negative, it is important to underline that there is no need to find an evident organic injury to support the diagnosis of pain as for psychosomatic pain or fibromyalgia; in these cases, a multidisciplinary approach is fundamental to reach a correct diagnosis [19]. Appendix A shows the cited articles of this paragraph.

### 3.2. Assessment

The assessment of pain is a critical step to provide adequate pain management and the lack of a standardized approach is one of the most problematic barriers to establish a tailored treatment [20]. 

The Initiative on Methods, Measurement, and Pain Assessment in Clinical Trials (IMMPACT) recommended to investigate six domains [21]: (i) pain; (ii) physical functioning; (iii) emotional functioning; (iv) patient ratings of improvement and satisfaction with treatment; (v) other symptoms and adverse events associated with past treatments; (vi) patient’s disposition and characteristics data.

Pain can be classified according to the underlying pathophysiological mechanism: in fact, it can arise from the stimulation of nociceptors in response to inflammation or damage (nociceptive pain), or from an injury or a disease of the peripheral or central nervous system (neuropathic pain). Recently, a definition of nociplastic pain has been introduced, defined as “pain that arises from altered nociception despite no clear evidence of actual or threatened tissue damage” [22]. 

During a patient’s assessment, a general medical history often reveals important co-factors, such as job, socioeconomic status, prior trauma, mood disorders, social life, as well as previous and current exposures to psychological and physical stimuli. It is recommended to report age, body weight, comorbidities, genetic status, and previous exposure to analgesic medication. 

The second step is to investigate the four W of pain assessment:-Where: the localization of pain;-When: the moment the pain started and the periodicity and frequency of its worsening;-What: what are the sensations (e.g., burning, stabbing, itching);-Why: possible causes of the pain that the patient can identify (e.g., trauma, surgery, job).

Physical examination in chronic pain relies on the different parts of the body involved in pain symptomatology and implies static and dynamic tests. For instance, in the case of low back pain, it is crucial to study the physio-pathological curves of the spine, the pelvic asymmetry, trigger points due to paravertebral muscle contractions, and pain induced by digital palpation of the spine [4,23,24]. Dynamic tests for low back pain comprehend Lasegue and Wasserman maneuvers, different tests for the elicitation of sacroiliac joint pain (e.g., distraction test, the posterior slipping, the FABER maneuver, the compression test, and the Gaenslen’s test), and gait evaluation. 

Even the assessment of chronic shoulder pain is traditionally based on a great variety of clinical tests which try to distinguish the origin of the pain among all the complex musculoskeletal structures of the joint: Hawkins’ and Neer’s tests for impingement, the empty-can test for supraspinatus injuries, the external rotation test for infraspinatus injuries, and the lift-off test for subscapularis lesions are only a few of all the provocative tests investigating the causes of shoulder pain [25]. 

Physical examination alone is not generally sufficient to identify the causes of pain because all the clinical tests have limited, and variable sensitivity and specificity and their results can often be operator-dependent: the correlation between physical exam, symptoms, and instrumental findings is pivotal to reach a correct diagnosis.

Multiple scales are available for assessing pain. The unidimensional scales only measure the intensity of the pain while the multidimensional scales evaluate location, intensity, psychological status, everyday functioning, and socialization. They are commonly used in clinical research, but they could help physicians in daily practice to reach a global assessment of the patient. 

The Numerical Rating Scale (NRS) is a very simple tool and it is considered the gold standard scale to assess pain [26]. A recent study demonstrated that the Visual Analogue Scale (VAS) and NRS are equally valid in assessing acute pain, while, on the contrary, the verbal rating scale (VRS) seems to underestimate its intensity. The Faces Pain Scale (FPS) is helpful in patients with reduced communicative skills (e.g., children, patients with cognitive or/and language impairment) [4,23,24]. 

The Brief Pain Inventory (BPI) is a well-known multiple item pain assessment instrument. It analyzes the severity and the degree of interference of pain in the patient’s life [27].

About pain quality, the McGill Pain Questionnaire (MPQ) investigates sensory and affective components of pain [27]. The Pain Quality Assessment Scale (PQAS) is useful to differentiate between nociceptive and neuropathic pain along with Douleur Neuropathique 4 (DN4) questions, which is a validated French questionnaire to evaluate neuropathic pain [28]. Appendix A reports the included articles of this paragraph.

### 3.3. Tailored Approach

In 1986, the World Health Organization developed a three-step ladder for cancer pain in order to suggest an escalation in the pharmacological treatment from non-steroidal anti-inflammatory drugs (NSAIDs) (first step) to weak (second step) and strong opioids (third step) based on pain intensity [29]. This first attempt to identify a tailored approach has not proved to be effective and many authors underlined that the original version of the ladder should be modified, eliminating the second step, incorporating a fourth step of minimally invasive procedures, and adopting a dynamic model moving through the four steps and mixing the approaches according to the patient’s needs [30]. 

Pain treatments include pharmacological, physical, behavioral, minimally invasive, and surgical interventions. Pharmacological therapy should take into account patient’s organ impairment, especially kidney and liver failure which are the primary sites of drugs’ metabolism and excretion; this factor generally implies a reduction in the dose of analgesics or a contraindication in their use [31]. It is important to mention that adrenal insufficiency and hypothyroidism can increase the effects of opioids and other analgesic drugs [32].

Physical and behavioral approaches could add a fundamental contribution in relieving pain, as many studies have underlined [33]. Minimally invasive and surgical interventions are generally reserved to refractory cases even if their advantages and drawbacks should be carefully balanced and compared to the adverse effects and efficacy of long-term pharmacological therapy (see the non-pharmacological approach section). Physicians must find the best approach to fit a patient’s needs, which include not only pain control, but also patient’s compliance; for example, some patients do not accept to have an implantable device even if it could be the right treatment for their pain as well as other patients who could not be able to manage a complex multidrug therapy.

Drug choice should be led primarily by the quality of pain: inflammatory pain would benefit from the use of NSAIDs, nociceptive pain from opioids, whereas neuropathic pain is generally treated with anticonvulsants and inhibitors of serotonin and noradrenaline reuptake [34]. It is difficult to identify a specific drug among a class which could adequately fit the patient’s needs and the choice is often based on the personal experience of the physician. Different molecules of the same class can have different effects, probably depending on receptor isoforms that could alter the pharmacological response: this is the concept underlying opioid rotation, i.e., the change of an opioid with another opioid due to adverse effects or ineffectiveness [35].

Contraindications and interactions [36] should be carefully taken into account; for instance, cyclooxygenase-2 (Cox-2) inhibitors should increase the risk of cardiovascular events, tricyclic antidepressants have been implicated in QT interval prolongation, opioids can cause respiratory depression [37]. Furthermore, many drugs used to treat neuropathic pain inhibit the reuptake on monoamines, so they should be carefully combined to avoid serious adverse reaction such as serotonin syndrome [38].

Treatment should always include not only drugs administered around the clock, but also rescue doses in the case of uncontrolled pain, which have to be prescribed according to the characteristics of the pain attack: for instance, breakthrough cancer pain usually needs drugs with a very rapid onset (5–10 min) with a 1–2 h action, whereas pain attacks both in malignant and benign conditions can arise more slowly and last longer [39,40].

Gene-tailored therapy is far from being a real alternative as pain is generally the result of a complex interaction between the nervous, immune, and endocrine system. Future tailored therapies include the development of new compounds and a more personalized approach based on individual genetic differences [41]. Appendix A summarizes the included articles of this paragraph.

### 3.4. Iterative Evaluation

An iterative assessment of the patient involves a periodical re-evaluation of pain intensity along with the assessment of treatment effectiveness. The timing of follow-up is based on the patient’s symptoms and therapy. Patients prescribed with opioid have to be assessed for the risk of addiction, while for patients who have recently started a new therapy it is fundamental to evaluate the efficacy of the treatment and its adverse effects [42]. COVID-19 lead to severe restrictions reducing patient access to the clinics, so the development of telehealth technology has been encouraged. Telemedicine can be very helpful in the follow-up, especially in patients who are stable or only need a few interventions to modify their current therapy [43]. Emerik et al. presented a clear description of the benefits and the appropriateness of telemedicine: one of the most important advantages is that patients living far from their reference hospitals or with disabling symptoms can easily be evaluated [44]. It is fundamental to establish which phases of patient management can be performed through telemedicine and when the traditional in-person visits are necessary; in fact, telemedicine could be inappropriate in patients with progressive symptoms, unclear diagnoses, complex medical and psychosocial conditions, and suspected drug abuse [45]. Appendix A illustrates the included articles of this paragraph.

### 3.5. Education

Patient education can be considered as the process through which healthcare providers give the patient both the information about his pain and the underlying condition and all the tools required to influence patient behavior and skills in order to effectively cope with the symptoms. The correct knowledge of the pain and of its management is fundamental, also because it can help the patient not feel trapped in his condition.

The patient has to be guided to recognize the factors that can influence pain such as sleep, nutrition, physical activity, mood, social life, flare-ups, and medications. Roberts MB et al. found a correlation between chronic pain and sleep disorders, while Brain K et al. showed that several nutrition interventions are effective in reducing pain [46,47]. Physical activity seems to decrease neural firing and increase endogenous opioid and serotonin levels in pain inhibitory pathways. Mental status and chronic pain can adversely affect each other, and they both can influence personal relationships. Moreover, patients have to be taught to recognize flare-ups and know the most suitable medications to treat them [48]. 

Many tools have been proposed to make patients more conscious of their condition: the simplest way is the use of brochures with all the basic information [49]. Video-assisted education can more efficiently engage patients; video–audio tools can be used in didactic presentation, with healthcare providers giving the information, practice presentation, showing real people engaged in an activity, and narrative presentation, including filming patients talking about their experience [50].

Cognitive–behavioral and educational interventions can improve both patients’ outcomes and family caregivers’ outcomes (e.g., adherence to therapy, fewer concerns about pain management, quality of life, the ability to recognize symptoms, flare-ups, worsening of background pain, and therapy efficacy) [51].

The caregivers’ education is equally important as they often are involved in medical and nursing tasks such as administration of therapy, monitoring of pain severity, medication effectiveness, and side effects; furthermore, caregivers have the role to communicate with the healthcare team and report all the information to manage pain therapy [51]. Similarly to patient education, the most adequate interventions for caregivers training is still a matter of debate. Appendix A illustrates the cited articles of this paragraph.

### 3.6. Non-Pharmacological Approach

Non-pharmacological approaches include invasive or minimally invasive treatments which are used to manage chronic pain refractory to traditional therapies; these treatments can dramatically reduce the dose of analgesic drugs and, consequently, their adverse effects. Even if these techniques could imply higher initial costs, in the long-term management they demonstrated to be economically advantageous both for the reduction in analgesic drugs and for a lower request of visits and hospital admissions [52]. The articles cited in this section are shown in Appendix A.

#### 3.6.1. Spinal Cord Stimulation 

Spinal Cord Stimulation (SCS) is the most used and successful electric neuromodulation approach to chronic pain. The exact mechanism of action of SCS is still undetermined and several theories have been so far advocated, including the gate-control theory, proposed by Melzack and Wall in 1965, and the involvement of wide dynamic range (WDR) neurons’ desensitization in the dorsal horn [53]. Spinal cord stimulation was initially characterized by the replacement of pain with paresthesia, whereas high-frequency 10 kHz (HF10) and burst stimulation (BS) are paresthesia-free; moreover, burst stimulation seems to modulate medial spino-thalamo-cortical pathways, which are responsible for the emotional and affective parts of painful sensation [54]. SCS probably acts not only through the inhibition of the nociceptive transmission and nociceptive neurons’ hyperactivity in the dorsal horn, but even through a supraspinal mechanism of stimulation of the inhibitory descending pathways [55].

Even if generally used in the case of pain refractory to pharmacological therapies, ever-growing evidence suggest the use of SCS as a first-line approach to some painful syndromes [56]. The most frequent indications are failed back surgery syndrome (FBSS), complex regional pain syndrome (CRPS), peripheral neuropathic pain, postherpetic neuralgia, intercostal neuralgia, refractory angina pectoris, phantom limb pain syndrome, epidural fibrosis, and cauda equina injury syndrome [57,58,59,60,61]. In FBSS, two RCTs have shown that tonic SCS was significantly superior to the best medical treatment alone and to repeated spine surgery, to alleviate lower limb pain, in terms of pain score (≥50% reduction), patient’s satisfaction, quality of life, and emotional impact [59,60]. In addition, the SENZA-RCT trial established the superiority and safety of HF10-SCS over conventional SCS in reducing chronic back and leg pain [62]. Similarly, in the SUNBURST study, BS-SCS met non-inferiority and superiority criteria compared to tonic stimulation for pain relief, patient’s quality of life, and safety profile [63]. 

The main contraindications are local infection or sepsis, coagulopathy, spinal stenosis, spina bifida, psychiatric disorders, somatization disorder, substance abuse, and patient’s refusal. Some relative contraindications are immune suppression and the presence of cardiac pacemaker or implanted defibrillators [57,58]. 

Appropriate patient selection by a multidisciplinary team is fundamental for the success of the implant: younger age, spinal cord stimulator tonic waveform, lower limb pain localization, and the absence of previous spine surgery positively influence the odds of successful SCS. Frequent complications are related to hardware dysfunction (e.g., migration, disconnection, electrode breakage) that may require additional surgery, or skin infection; skin infections have to be carefully managed as they can migrate along the lead and cause central nervous system infections such as meningitis. Exceptionally, spinal cord damage can occur, either in the form of direct lesions or of indirect compression by epidural hematoma [64]. 

#### 3.6.2. Radiofrequency

Radiofrequency (RF) stimulation is a safe and effective treatment to control different types of chronic neuropathic pain; it is performed through a needle percutaneously placed next to the target nerve under fluoroscopy, CT guidance, or ultrasound guidance. The continuous radiofrequency (CRF) technique uses a continuous electrical stimulation at a frequency of 400–500 kHz for 2 min, reaching a temperature of 60–90 °C with a consequent irreversible thermoablation of the target nerve. In contrast, pulsed radiofrequency (PRF) produces short heat bursts, with long resting phases between them in order to keep the temperature under the limit of protein denaturation of 42 °C, thus preventing irreversible nervous injuries [65]. PRF is supposed to have neuromodulatory and anti-inflammatory effects such as the decrease in microglia activity and the increase in c-fos expression in the dorsal horn; furthermore, it seems to potentiate the noradrenergic and serotonergic descending pain inhibitory pathways. C and A∂ fibers, which carry nociceptive stimuli from the periphery to the dorsal horn, seem to be microscopically damaged by PRF, in contrast with the Aβ fibers, which carry no pain-related signals and which are rarely involved [66,67].

Significant evidence supports RF stimulation as an effective and safe treatment for cervical and lumbar radicular and facet joint pain; in particular, the CRF of medial branch seems to be more effective and long-lasting than PRF with no motor damage risks.

Recently, PRF has emerged as a safe and potentially effective treatment also for postherpetic neuralgia and occipital neuralgia and it resulted as being superior to oral medication or epidural infusion of anesthetics; however, the CRF procedure should be avoided because it may induce several adverse effects, including sensory loss, dysesthesia, and anesthesia dolorosa [68,69,70]. The PRF of the nervus suprascapularis may relieve shoulder pain and can improve joint mobility, whereas the ablation of genicular nerves can be effectively used to treat knee pain [71]. The effectiveness of radiofrequency in other spinal disorders such as cervicogenic headaches, discogenic pain, thoracic facet joint pain, and coccydynia, is still uncertain; similarly, additional prospective clinical trials are necessary for clarifying the usefulness of PRF in meralgia paresthetica, carpal tunnel syndrome, tarsal tunnel syndrome, and Morton’s neuroma [19,65].

Cooled radiofrequency is a novel technique which uses a cooled needle to generate a larger ablation area: in fact, CRF, due to the high temperature reached around the needle, creates a charred layer which acts as an insulator, hindering the spreading of radiofrequency through the tissue [72].

Radiofrequency techniques are characterized by great potential applications, a high level of safety, low cost, reduction in analgesic drug use, and long-lasting effects, but, on the other hand, they are still not supported by consistent scientific evidence.

#### 3.6.3. Other Non-Pharmacological Approaches

Intrathecal drug delivery devices (IDDDs) consist of infusion systems to administer drugs into the subarachnoid space through a catheter connected to a subcutaneously implanted pump. The injection of analgesics into cerebrospinal fluid makes it possible to by-pass the blood–brain barrier and the administration of a lower dosage of drugs to obtain the same analgesic action compared to other routes with fewer side effects [73]. Drugs are initially administrated at the lowest dose, defined as the trailing dose, which can reach a pain reduction of 30–70% (usually 50%) and then be titrated according to side effects and patient response [74]. The drugs currently approved by the Food and Drug Administration (FDA) and European Medicines Agency (EMA) for intrathecal administration are morphine and ziconotide, but many others such as hydromorphone, fentanyl, bupivacaine, and clonidine are used off-label in clinical practice [74]. IDDDs may be effective and safe for patients suffering from refractory chronic pain, both malignant and benign such as compression fractures, spondylolisthesis, spondylosis, FBSS, spinal stenosis, CPRS, radicular pain, postherpetic neuralgia, and post-thoracotomy syndromes [75]. A multidisciplinary team should select eligible patients based on pain characteristics and intensity, quality of life, other medical treatments, associated comorbidities, and degree of disability along with pre-implantation psychological condition and, for malignant pain, cancer stage and grade [76]. Procedural-related complications, which should be rapidly identified and treated, are epidural hematoma, epidural abscess, sustained loss of CSF, spontaneous intracranial hypotension, postdural puncture headache, and device failure or displacement [58].

Chemical neurolysis acts through aggressive chemical compounds in order to destroy the target nerve; it is used to treat trigeminal neuralgia, especially in patients who refuse or are ineligible to microvascular decompression [77,78,79,80]. Despite their efficacy, alcohol and phenol have significant side effects and were replaced by glycerol. Xu et al. described pain relief in 73% of their 3370 patients after just one injection, with an overall success rate of 99.58% after four injections [81]. However, 20–40% of the patients experienced pain recurrence within a median of two-five years after the procedure [81]. Another ablative approach is cryoablation, which causes axonotmesis of the affected nerve using temperatures between −60 °C and −140 °C [82]. However, due to overall shorter pain-free duration (6–9 months in most cases) compared with other percutaneous techniques, it is not a first-line treatment of trigeminal neuralgia [82]. 

Acupuncture is a controversial treatment for chronic pain, due to its non-biomedical origin. Traditionally, the insertion of needles in specific points of the body is thought to restore the normal flow of energy [83]. In the modern era, its exact mechanism of action is still unknown: acupuncture seems to lead to long-term decreases in pain with multiple effects on the central and peripheral nervous systems. Functional magnetic resonance imaging has shown the involvement of neuron clusters releasing endogenous opioids, serotonin, and norepinephrine, which may have downstream effects on nociceptors, inflammatory cytokines, and other physiologic pain pathways [83]. According to a recent meta-analysis, verum acupuncture is significantly superior to sham control, suggesting that its effectiveness is not merely related to a placebo effect [84]. Although recommendations are generally not strong and based on a low or moderate quality of evidence, acupuncture has been shown to be helpful in some pain syndromes such as low back pain, knee osteoarthritis, headaches, migraines, myofascial syndromes, and fibromyalgia [85,86,87].

Regular, long-term, physical exercise is always helpful to achieve pain relief. Resistance-based strengthening and stabilization exercises have significantly improved patient-reported pain in older individuals with a primary diagnosis of hip or knee osteoarthritis and degenerative lumbar spondylosis [88,89]. Additionally, low-impact modalities such as tai chi, yoga and aqua-aerobic regimens may modestly improve balance and musculoskeletal function when performed regularly [90].

Psychological aspects of pain must be treated. Chronic pain patients tend to feel depressed, anxious, and lonely. Psychotherapy, music therapy, and pet therapy can be helpful in relieving pain, anxiety, fatigue, and improving quality of life [91].

### 3.7. Team

The multi-dimensional aspect of pain suggests that its optimal assessment and management may be best achieved using a multidisciplinary approach [92]. Unfortunately, pain specialists are often not involved in patient management early enough, whereas they should have a central coordinating role in the pain team. The pain team should be composed of different figures according to patient pain characteristics and comorbidity, such as oncologists, clinical psychologists, physiotherapists, geriatricians, rheumatologists, orthopedic surgeons, neurosurgeons, and nutritionists. The multidisciplinary approach must involve pharmacological, invasive and mini-invasive interventions, physical rehabilitation, cognitive–behavioral and occupational therapy, and psychological interventions to manage depression and anxiety.

Older patients are at high risk for polypharmacy and medication mismanagement and require coordination between physicians and patients’ caregivers or long-term care facilities [93,94]. The aim is to offer a treatment that is tailored on patient-specific needs and to maximize the degree of satisfaction and quality of life while minimizing the risks of adverse psycho-physical consequences.

Telemedicine is a useful tool for healthcare providers to keep in touch, thus favoring the collaboration between them and the coordination of interdisciplinary interventions.

A team-based approach is associated with significant improvements in pain intensity, daily mood, disability, and quality of life [92]. Interdisciplinary approaches can be based on the simultaneous or stepped combination of the above-mentioned specialists and treatments, and both approaches have proven to guarantee a better management of chronic pain [95,96]. Appendix A reports the evaluated articles of this section.

## 4. Conclusions and Future Directions

Our narrative review represents, to the best of our knowledge, the first attempt to summarize all the main available data about the steps to follow in the management of chronic pain, gathering them in a simple acronym. Pain is one of the most important challenges for every physician starting from its assessment, which is jeopardized by the different way a patient experiences and reports his symptoms, up to the treatment, which can have different results on the basis of the patient’s characteristics; it is equally important to know all the available pharmacological and minimally invasive treatments to choose the best therapy which should be thoroughly tailored to the patient. Moreover, ever-growing evidence supports the importance of a multidisciplinary approach to better respond to the complexity of pain and the essential aspect of cooperation among different specialists to achieve therapeutic success.

## Figures and Tables

**Figure 1 jpm-13-01551-f001:**
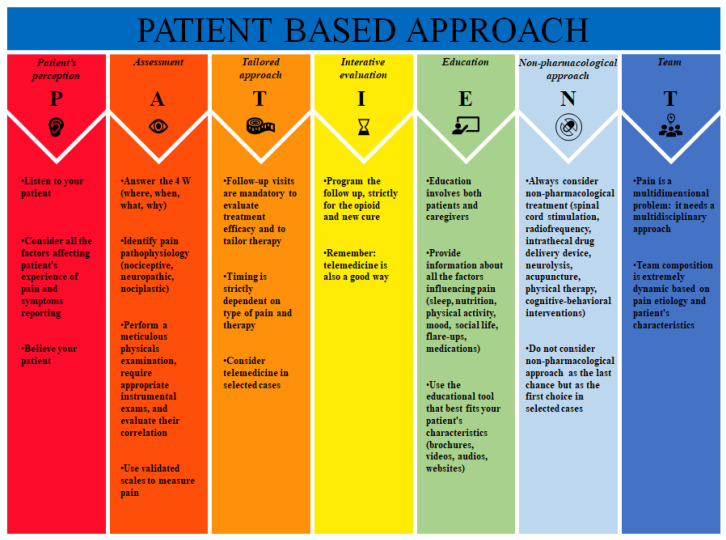
PATIENT bundle.

## Data Availability

No data were collected to write this manuscript. All used references are reported in the References.

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
