# Peer review of "The PATIENT Approach: A New Bundle for the Management of Chronic Pain"

_jpm, 2023, doi:10.3390/jpm13111551_

Round 1

Reviewer 1 Report

Comments and Suggestions for Authors

Dear Author,

The topic of the review is very important since chronic pain is a crucial global health problem. However, there are several published reviews on this aspect. Therefore, the originality of this compilation should be emphasized. Is the term "PATIENT" invented by the authors? If it is an original definition, this part should be emphasized and explained in detail.

As stated in the method section, only Pubmed was used as a source in the preparation of the review. The years in which the article search was made should be written in the methods section of the main manuscript and abstract.

There are some spelling, grammar, and punctuation errors in the manuscript. The manuscript should be checked by a native speaker or a grammar program.

Comments on the Quality of English Language

There are some spelling, grammar, and punctuation errors in the manuscript.

Author Response

Reviewer 1

Comments and Suggestions for Authors

Dear Author,

The topic of the review is very important since chronic pain is a crucial global health problem. However, there are several published reviews on this aspect. Therefore, the originality of this compilation should be emphasized. Is the term "PATIENT" invented by the authors? If it is an original definition, this part should be emphasized and explained in detail.

We underlined in the introduction and discussion that the acronym PATIENT was invented by us and that this is the first narrative review to our knowledge that tried to summarize all the different aspects of pain management in an acronym.

As stated in the method section, only Pubmed was used as a source in the preparation of the review. The years in which the article search was made should be written in the methods section of the main manuscript and abstract.

We specified in the methods section the range of time in which the articles for our review were searched.

 There are some spelling, grammar, and punctuation errors in the manuscript. The manuscript should be checked by a native speaker or a grammar program.

We performed a revision of the entire article to correct spelling, grammar, and punctuation errors.

Reviewer 2 Report

Comments and Suggestions for Authors

Dear Author. This article has an interesting title. I hope it can be published after the following major modifications:

1- Please, in the introduction of the article, more mention should be made of the importance and necessity of conducting this research and the GAP in the previous sources that conveyed the necessity of this research to the minds of current researchers.

2- Why was the search for articles only done in PubMed and other search engines were not used to find ISI and Scopus articles?

3- In the section on materials and methods of work, lines 55 to 74, a large number of words cause the reader's mind to break apart. It is recommended to mention only the search strategy, the year of the study, and the number of years that the articles cover.

4- Please explain exactly how many articles were found in the initial search and under what conditions were they reviewed or rejected in the study.

5- At the end, how many articles were reviewed in each item?

6- Despite the fact that I acknowledge its usefulness, the way of writing is more like a pamphlet. Please prepare a table in each item to review the articles of the same section, so that it resembles the format of a review article.

7- In the section Non-pharmacological approach, from line 259 onwards, why are only three approaches selected and examined while numerous therapeutic modalities and exercises are used in physiotherapy to reduce pain?

8- It is suggested that items such as the list of complete words that were used for searching, tables, and graphs summarizing the articles in each examined item should be included in the supplementary information.

Author Response

Reviewer 2

Dear Author. This article has an interesting title. I hope it can be published after the following major modifications:

1- Please, in the introduction of the article, more mention should be made of the importance and necessity of conducting this research and the GAP in the previous sources that conveyed the necessity of this research to the minds of current researchers.

We specified in the introduction that our acronym is new and it was coined both to help clinicians remember the steps to follow in the management of chronic pain, and to gather the available evidences for every topic which the letters of the acronym stand for; in fact, the articles published so far focused their attention on specific aspects of chronic pain with no works trying to put together all the information in a comprehensive view.

2- Why was the search for articles only done in PubMed and other search engines were not used to find ISI and Scopus articles?

The articles we found in ISI and Scopus (and Cochrane database) search were all included in PubMed search results and this is the reason why we cited only PubMed. As the reviewer suggested, it is more correct to clarify that the search was done in all the above-mentioned database.

3- In the section on materials and methods of work, lines 55 to 74, a large number of words cause the reader's mind to break apart. It is recommended to mention only the search strategy, the year of the study, and the number of years that the articles cover.

Thanks to the reviewer for this suggestion. Actually, this part is very difficult to read, so we simply clarify the search strategy in the main text of the manuscript along with the range of time in which the articles were searched. We moved in the supplementary material the terms and their combinations used for the database search.

4- Please explain exactly how many articles were found in the initial search and under what conditions were they reviewed or rejected in the study.

We reported in the table S1 of supplementary material the number of articles we initially found and the number of articles we included in the review for every topic

Regarding inclusion and exclusion criteria, as we performed a narrative review, we did not embrace strict and stringent inclusion and exclusion criteria for the article selection. For every item, we excluded the articles which do not specifically address the topic of the section, firstly from the title and then by reading the abstract and the full text; for the research articles addressing the same issue, we chose the most comprehensive work with the more rigorous approach in terms of sample size, study design, and statistical analysis, preferring randomized controlled trial over retrospective, non-randomised, or non-controlled trials; in case of meta-analysis, we included articles with adequate consideration of risk of bias, accurate statistical approach and stringent selection criteria of included articles; for the reviews we selected the most comprehensive work on every specific item.

The main aim of our work was to create an acronym to make clinicians keep in mind the steps of chronic pain management; starting from that purpose, we review the literature to convey the main information about every topic we treated.

5- At the end, how many articles were reviewed in each item?

As we wrote above, Table S1 reports the strings for the query related to every topic; furthermore, the table reports the number of articles initially found and the number of articles eventually included in the review.

6- Despite the fact that I acknowledge its usefulness, the way of writing is more like a pamphlet. Please prepare a table in each item to review the articles of the same section, so that it resembles the format of a review article.

We included in the supplementary material a table for every item of the acronym PATIENT in which we summarize the findings of every article we included in the review (Table S2-S3-S4-S5-S6-S7-S8).

7- In the section Non-pharmacological approach, from line 259 onwards, why are only three approaches selected and examined while numerous therapeutic modalities and exercises are used in physiotherapy to reduce pain?

The section of non-pharmacological therapy wanted to give a comprehensive view of all the approaches other than drugs that clinicians should consider: we cited the physical therapy as a possible alternative or a complementary treatment, but a deep description of all the possible modalities and exercises which physiotherapy uses to reduce pain would have excessively lengthened the section, going beyond the intentions of the review which would like to give simple and direct suggestions about what to consider in the management of chronic pain, reserving the discussion and the decision of the specific modalities of physiotherapy to the specialist.

8- It is suggested that items such as the list of complete words that were used for searching, tables, and graphs summarizing the articles in each examined item should be included in the supplementary information.

We included a table with the search terms and query strings for every item as suggested by the reviewer.